# Comparing the Efficacy of OnabotulinumtoxinA, Sacral Neuromodulation, and Peripheral Tibial Nerve Stimulation as Third Line Treatment for the Management of Overactive Bladder Symptoms in Adults: Systematic Review and Network Meta-Analysis

**DOI:** 10.3390/toxins12020128

**Published:** 2020-02-18

**Authors:** Chi-Wen Lo, Mei-Yi Wu, Stephen Shei-Dei Yang, Fu-Shan Jaw, Shang-Jen Chang

**Affiliations:** 1Institute of Biomedical Engineering, National Taiwan University, Taipei 10617, Taiwan; 2Division of Urology, Department of Surgery, Taipei Tzu Chi Hospital, The Buddhist Tzu Chi Medical Foundation, New Taipei 23142, Taiwan; 3School of Medicine, Buddhist Tzu Chi University, Hualien 97071, Taiwan; 4Department of Nephrology, Taipei Medical University-Shuang Ho Hospital, Taipei 23561, Taiwan

**Keywords:** network meta-analysis, OnabotulinumtoxinA, overactive bladder, peripheral tibial nerve stimulation, sacral neuromodulation

## Abstract

The American Urological Association guidelines for the management of non-neurogenic overactive bladder (OAB) recommend the use of OnabotulinumtoxinA, sacral neuromodulation (SNM), and peripheral tibial nerve stimulation (PTNS) as third line treatment options with no treatment hierarchy. The current study used network meta-analysis to compare the efficacy of these three modalities for managing adult OAB syndrome. We performed systematic literature searches of several databases from January 1995 to September 2019 with language restricted to English. All randomized control trials that compared any dose of OnabotulinumtoxinA, SNM, and PTNS with each other or a placebo for the management of adult OAB were included in the study. Overall, 17 randomized control trials, with a follow up of 3–6 months in the predominance of trials (range 1.5–24 months), were included for analysis. For each trial outcome, the results were reported as an average number of episodes of the outcome at baseline. Compared with the placebo, all three treatments were more efficacious for the selected outcome parameters. OnabotulinumtoxinA resulted in a higher number of complications, including urinary tract infection and urine retention. Compared with OnabotulinumtoxinA and PTNS, SNM resulted in the greatest reduction in urinary incontinence episodes and voiding frequency. However, comparison of their long-term efficacy was lacking. Further studies on the long-term effectiveness of the three treatment options, with standardized questionnaires and parameters are warranted.

## 1. Introduction

Overactive bladder (OAB) syndrome is defined as “the presence of urinary urgency, usually accompanied by frequency and nocturia, with or without urgency urinary incontinence, in the absence of urinary tract infection (UTI) or other obvious pathology” [1]. The prevalence of OAB syndrome increases with age and there is no significant gender difference [2]. Non-neurogenic OAB impairs the patient’s quality of life (QoL) and behavioral therapy is recommended as the first line treatment. If behavioral therapy fails, oral medications, including antimuscarinics and β3 agonists, are recommended as the second line therapy [3]. When there is inadequate symptom control or intolerable side effects due to second line management, the American Urological Association (AUA) guidelines recommend either OnabotulinumtoxinA, sacral neuromodulation (SNM), or peripheral tibial nerve stimulation (PTNS) as third line therapy options for OAB symptoms. Third line therapy is undertaken if the patient desires further treatment and is willing to engage in treatment, and/or further treatment is determined by clinicians to be in the patient’s best interests. At present, the decision on which third line therapy to perform is based on the clinicians’ and patient’s preference, and there is not an evidence-based hierarchy available for guidance [3].

There have been several previously published randomized control studies, which compared pairwise treatments with a placebo [4,5]. However, there has not been a direct comparison of the three available treatments, and there has also been a lack of efficiency and safety comparisons between the three treatment options. When multiple treatment modalities are considered, a network meta-analysis could help compare their efficacies. Therefore, we conducted a systemic review to compare the efficacy of OnabotulinumtoxinA, SNM, and PTNS for the treatment of OAB symptoms, using a network meta-analysis.

## 2. Results

### 2.1. Included Studies

A Preferred reporting items for systematic reviews and meta-analyses (PRISMA) flow diagram flowchart summarizing the literature search is shown in Figure 1 [6]. The initial search identified 1940 and 5722 potential studies from PubMed and EMBASE, respectively. After the removal of duplicates the total number of articles was 7662. After screening, a total of 5738 articles were excluded based on their title and/or abstract, while another 185 articles were removed after a full-text assessment. A total of 20 articles met the qualitative inclusion criteria, while 17 trials, including 3038 participants, met the criteria for systematic review and network meta-analysis.

### 2.2. Study and Participant Characteristics

The number of patients, the study design, and the inclusion and exclusion criteria for each of the included studies are listed in Table 1. As the three investigated treatment modalities are used for third line OAB syndromes management, most of the included patients were refractory or intolerant to the first and second line treatments. 

### 2.3. Networks

There was sufficient evidence available for analysis of the following efficacy and safety endpoints: urinary frequency per day, incontinence episodes per day, ≥50% reduction of symptoms, patients with urinary tract infections (UTIs), and post-treatment urine retention needing clean intermittent catheterization (CIC). There was a lack of sufficient data to make comparisons between the three treatment modalities with regard to the QoL, urgency, urge incontinence episodes/day, maximal bladder capacity, and nocturia. Results of the pair-wise comparison meta-analyses are shown in Table 2.

### 2.4. Risk of Bias Assessment

A summary of the included studies and a risk of bias graph are shown in Figure 2. The four studies that compared SNM with delayed SNM were rated as having a high risk of bias in the ‘measurement of outcome’ category because the self-reporting results could have been influenced by the placebo effect. As these papers did not describe the randomization method or specify whether the assessors were blinded, we had some concerns regarding the randomization process when the risk of bias was evaluated. All of the studies, except the four that compared the efficacy of SNM with the delayed SNM group, were judged as having a ‘low risk’ of bias or as having ‘some concerns’.

### 2.5. Network Meta-Analysis on the Outcomes of Interests

#### 2.5.1. Efficacy

##### Urinary Frequency per Day

A total of nine studies contributed to the comparison of urine frequency per day [6,7,8,9,10,11,12,13,14]. Pair-wise comparisons with a random effects (RE) model revealed that all three modalities were more efficacious than the placebo (Table 2). The NMA (Network Meta-analysis) identified a greater reduction in the total number of micturition per day for SNM compared with the placebo, PTNS, and OnabotulinumtoxinA. There were no significant differences observed between OnabotulinumtoxinA and SNM (Table 3). The ranking probability results are shown in Figure 3. The ranking results for urinary frequency reduction was as follows: SNM ranked first, OnabotulinumtoxinA ranked second, PTNS ranked third, and placebo ranked fourth (Figure 3A).

##### Urinary Incontinence Episodes per Day

There were seven studies used to compare the efficacy of the three modalities on the number of incontinence episodes per day at 12 weeks follow-up [12,13,15,16,17,20,22]. Pair-wise comparisons with a RE model revealed that all three modalities were more efficacious than the placebo (Table 2). The NMA demonstrated that SNM was associated with a greater reduction in the total number of incontinence episodes per day compared with the placebo, PTNS, and OnabotulinumtoxinA. There was no significant difference between the efficacy of OnabotulinumtoxinA and PTNS (Table 3). The ranking results for incontinence episode reduction was as follows: SNM ranked first, PTNS ranked second, OnabotulinumtoxinA ranked third, and placebo ranked fourth (Figure 3B).

##### ≥50% Symptom Improvement at 12 Weeks Follow-up

The network of eligible comparisons for >50% symptom improvement is shown in Figure 4D. There were eight studies that reported parameters including ≥50% improvement in symptoms at 12 weeks follow-up. Pair-wise comparisons with a RE model revealed all three modalities were more efficacious than the placebo. However, there was a significant inconsistency expected in ≥50% reduction of symptoms improvement. Therefore, network meta-analysis and ranking probability was not conducted with regard to this parameter.

### 2.6. Complications

#### 2.6.1. Urinary Tract Infection

Estimates of the treatment effectiveness on the occurrence of UTIs were informed by 10 studies [7,8,9,11,12,13,14,15,16,23,24]. The RE model revealed that OnabotulinumtoxinA was associated with a higher incidence of UTIs compared with the placebo, SNM, and PTNS (Table 3). The ranking results for the post-management risk of UTIs was as follows: PTNS ranked best, SNM ranked second, placebo ranked third, and OnabotulinumtoxinA ranked worst (Figure 3C).

#### 2.6.2. Urine Retention Needing Clean Intermittent Catheterization

Estimates on the effect of treatments on post-management urine retention were reported in 11 studies [7,8,9,10,11,12,13,14,15,16,23,24]. The RE model demonstrated that OnabotulinumtoxinA was associated with a higher occurrence of post-treatment urine retention needing catherization compared with the placebo, SNM, and PTNS (Table 3). The ranking results for post-management risk of post-treatment urine retention was as follows: SNM ranked best, placebo ranked second, PTNS ranked third, and OnabotulinumtoxinA ranked worst (Figure 3D).

## 3. Discussion

The AUA guidelines suggest intra-detrusor injection with OnabotulinumtoxinA (evidence grade B), SNM (evidence grade C), or PTNS (evidence grade C) as the third-line treatment modalities for adult OAB [3]. In the absence of a direct head to head comparison of the three treatment modalities for adult OAB symptoms, the present detailed systemic review and network meta-analysis is the first review to combine all updated evidence and compares the efficacy of any dose of OnabotulinumtoxinA, sacral neuromodulation and PTNS. The compared outcomes included voiding frequency/day, urinary incontinence episodes/day, and ≥50% reduction in symptoms. Pairwise meta-analysis revealed that all three modalities were more efficacious than a placebo with regard to the outcomes of interests, including urinary frequency, incontinence, and achieving ≥50% of symptoms improvement. SNM achieved the greatest reduction in urinary incontinence episodes and voiding frequency/day. OnabotulinumtoxinA was associated with the highest risk of urine retention and UTI episodes in the follow-up period. As none of the included studies used a unified or standard questionnaire to evaluate the QoL, the results regarding QoL were not pooled for the meta-analysis. We suggested that International Continence Society or International Urogynecology Association should unify the QoL questionnaire based on evidence and experts’ opinion for a better evaluation of post treatment result.

The cost of the treatment and the insurance payment system may influence the patient’s preference for a specific therapy. A cost analysis was performed to assess the economic effectiveness of each treatment. Based on a literature review, SNM was considered to be the most expensive treatment compared with a OnabotulinumtoxinA injection and PTNS in the short term [25,26,27]. However, in a model of middle- and long-term treatment, the cost-effectiveness of SNM was comparable with OnabotulinumtoxinA [25,28]. There was no comparison between OnabotulinumtoxinA and PTNS. Martinson et al. constructed the Markov model to simulate the cost-effectiveness of PTNS and they concluded that PTNS was the least costly therapy compared with OnabotulinumtoxinA and SNM [26]. However, different regional health insurance and health care payment systems could affect the simulation result and lead to different outcomes. A local cost-effectiveness analysis is more valuable for urologists.

The current study primarily compared short-term efficacy at 12 weeks follow-up and data comparing long-term efficacy is lacking. Amundsen et al. conducted a randomized control trial with a 6 years follow-up, which confirmed the middle- to long-term efficacy, QoL and satisfaction with treatment for OnabotulinumtoxinA injection and SNM [24]. However, there has been no previous report on the long-term efficacy of PTNS. The present study did not compare long-term adherence between treatment modalities.

The major drawbacks for management with antimuscarinics lie in the low rates of adherence to the medication [29,30]. Adherence at 12 months was 39% for mirabegron vs. 14–35% for antimuscarinics [29]. The effects of OnabotulinumtoxinA persist for 6–9 months and the effect can be extended with repeated injections [31]. Long-term adherence to OnabotulinumtoxinA injections is less discussed. Patient preference is an important factor, especially when comprehensive data is not available to assist in the decision-making progress. No significant difference in patient satisfaction was reported between SNM and OnabotulinumtoxinA in the ROSETTA trials [23,24] or studies by Hoag et al. [32]. Since all three of the investigated third line treatments are effective, health care providers should carefully discuss the pros and cons of each treatment with the patient and determine the appropriate strategy based on each individual situation.

The adverse effects of an OnabotulinumtoxinA injection include hematuria, bacteriuria, UTIs, urine retention, and increased post-void residual urine [33]. The rate of post-therapeutic complications, including urine retention that needed clean intermittent catheterization and UTIs, were compared between treatments. Intravesical injection with OnabotulinumtoxinA lead to a significantly higher rate of CIC and UTIs. Nevertheless, the higher rate of CIC was not consistent among the included trials and the literature review, therefore this conclusion could be controversial. Side effects associated with SNM included pain at the stimulator and lead sites, lead migration, infection, and the requirement for surgical revision [21,22,23,24,32]. A screen test before implantation and two-stage implantation could increase the success rate [34]. The revision rate varied from 3–32% and the removal rate varied from 8.6–13% [24,35]. With improvements in battery longevity and better localization of lead placement, these revision and removal rates could be reduced. The only side effects of PTNS were local adverse events, including minor bleeding spots and temporary pain [4].

There were several limitations to the current network meta-analysis study. First, the variable qualities and publication biases of the included studies may compromise the results of the network meta-analysis. There was considerable heterogeneity across the study designs, including participants, scheduled follow-ups, questionnaires, evaluated parameters, OnabotulinumtoxinA dose, and SNM and PTNS protocols. The U.S Food and Drug Administration approved intra-detrusor injection of OnabotulinumtoxinA is 100 U [3], while the included ROSETTA trial used 200 U [23,24]. Different protocols were also used for PTNS. Second, the variable or ambiguous and potentially post-hoc definitions for the outcome parameters used in each study made it difficult to unify and compare the treatment results between the studies. However, because of the heterogeneity of enrolled patients, existing of high-risk bias and lacking long-term outcomes comparison, more high-quality studies are necessary to clarify this benefit and risk of the current available 3rd line therapy for overactive bladder symptoms. A strength of the current study was that it collected 17 randomized controlled trials published in English over three decades and used network meta-analysis to compare the three existing therapeutic options.

## 4. Conclusions

The results revealed that all three modalities were efficacious in managing adult OAB syndrome, and all were better than a placebo on the specific symptoms reported to be the outcome of the study. This review shows that at 12 weeks follow-up, SNM yielded the greatest reduction in urinary incontinence episodes and urinary frequency/day. OnabotulinumtoxinA resulted in a higher incidence of complications, including urinary tract infection and urinary retention.

As there is a lack of head to head comparison studies among SNM, PTNS, and OnabotulinumtoxinA for the treatment of adult OAB symptoms, the current network meta-analysis represents the best available evidence for the comparison of these three treatment modalities.

## 5. Material and Methods

### 5.1. Search Strategy and Eligibility Criteria

To identify appropriate studies for a network meta-analysis, two independent investigators (CWL and SJC) conducted a comprehensive electronic literature search of PubMed, EMBASE, Cochrane Central Register of Controlled Trials (CENTRAL), Wiley, and ClinicalTrial.gov for trials published between January 1995 and September 2019, with a language restriction of English. The present study followed PRISMA recommendations. The terms and related synonyms “overactive bladder”, “bladder overactivity”, “detrusor overactivity”, “urinary incontinence”, “urgency”, “urgent incontinence”, “detrusor overactivity” and “Botulinum A toxin” or “OnabotulinumtoxinA” or “Botox” or “botulinumtoxin A” or “sacral neuromodulation” or “sacral nerve stimulation” or “percutaneous tibial nerve stimulation” or “posterior tibial nerve stimulation” were combined in the search strategy. MeSH terms, key words and other free terms were used for searching and Boolean operators (OR and AND) were used to combine the searches. The reference lists of the included articles, as well as the guidelines for the AUA and European Association of Urology were manually reviewed, and external peer reviewers were asked to contribute any additional trials.

Studies were considered eligible if they were randomized controlled trials that compared any dose of OnabotulinumtoxinA, SNM, and PTNS therapy with each other or a placebo, in adults with OAB syndrome with reported efficacy at 12 weeks follow-up. It was required that the studies provided detailed data on the treatments and outcomes of the participants.

### 5.2. Data Extraction and Quality Asessment

Two investigators (CWL and SJC) independently reviewed the titles and abstracts to check their relevance and adherence to the eligibility criteria. The full text of articles was assessed if their eligibility was not clear from the abstract. A preliminary network was constructed based on the intervention and comparators in the included trials. The homogeneity of the included trials was also evaluated.

Two investigators (CWL and SJC) reviewed the quality of the included studies. The risk of bias was evaluated using the RoB 2 (Version 2 of the Cochrane risk-of-bias tool for randomized trials) tool to evaluate the quality of evidence [36]. The following domains were evaluated: random sequence generation, allocation concealment, blinding, incomplete data on outcomes, selective reporting, and other bias. The Grading of Recommendation, Assessment, Development, and Evaluation Working Group approach for rating the quality of treatment effect estimates was also followed. Urination frequency/day, incontinence, and ≥50% symptom improvement were selected because of their relevance to clinical symptom improvement. The quality of evidence was rated as very low, low, moderate or high.

When standard deviation data was missing or only 95% confidence intervals (CI) were listed, the standard deviation was calculated using the formula in the Cochrane handbook for systematic reviews of interventions, or it was calculated from the figure data in the article or supplemental data.

### 5.3. Network Development

If the data was available, theoretical networks were developed for each outcome based on the similarities between studies. The outcomes included QoL, lower urinary tract symptoms at 12 weeks follow-up, including urinary incontinence episodes, urgency episodes, urge urinary incontinence episodes/day and urinary frequency/day, achieving ≥50% of symptoms improvement, nocturia, and complications including post-treatment urine retention that needed intermittent catheterization and urinary tract infections.

### 5.4. Statistical Analysis

The network meta-analysis was performed using R 3.3.2 software (Bell Laboratories, Madison, WI, USA, 2016) and STATA version 15.0 (StataCorp. LP, College Station, TX, USA, 2017). All outcomes of interests were compared pairwise by calculating *I^2^* statistics. Study heterogeneity was assessed using the R package. Node splitting analysis was performed to evaluate inconsistencies by comparing differences between the direct and indirect evidence. Dichotomous variables and continuous variables were expressed as odds ratios (ORs) with 95% CIs and weighted mean differences with 95% CIs, respectively. A RE model was used to calculate evidence inconsistencies because of the existence of heterogeneity among the included trials and each intervention comparison. The ranking probabilities for the different OAB symptoms interventions were also calculated with regard to each outcome of interest. Additionally, publication bias was evaluated according to the symmetry characteristics of funnel plots, with a symmetrical and concentrated distribution of dots implying no significant deviation.

## Figures and Tables

**Figure 1 toxins-12-00128-f001:**
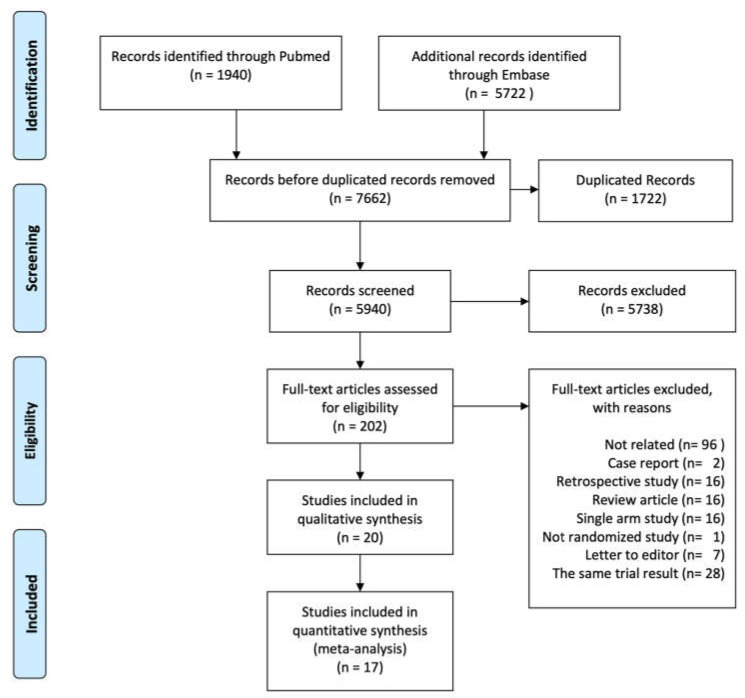
Preferred reporting items for systematic reviews and meta-analyses (PRISMA) flow diagram of the study selection process for network meta-analysis. The figure was generated using the PRISMA 2009 Flow Diagram.

**Figure 2 toxins-12-00128-f002:**
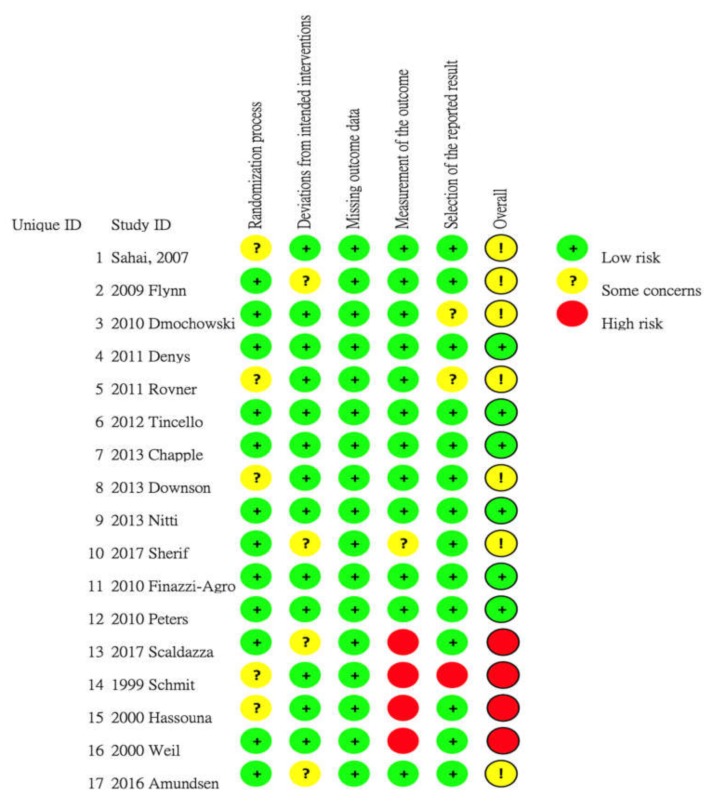
Risk of bias graph and summary of the included studies: Reviewers’ judgments regarding each risk of bias item for the included studies. The figure was generated using RoB 2 tool (the 22 August 2019 version)

**Figure 3 toxins-12-00128-f003:**
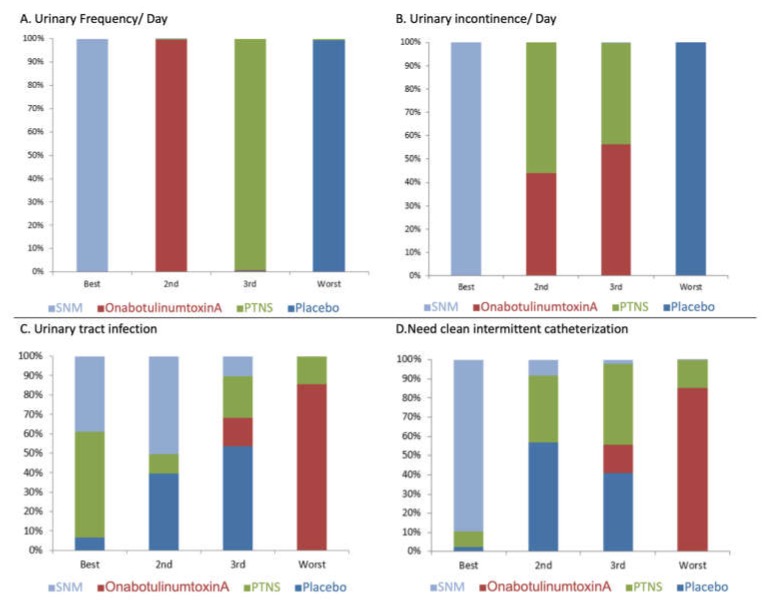
Treatment rankings for (**A**) urinary frequency/day, (**B**) incontinence, (**C**) urinary tract infection, and (**D**) urine retention needing clean intermittent catheterization.

**Figure 4 toxins-12-00128-f004:**
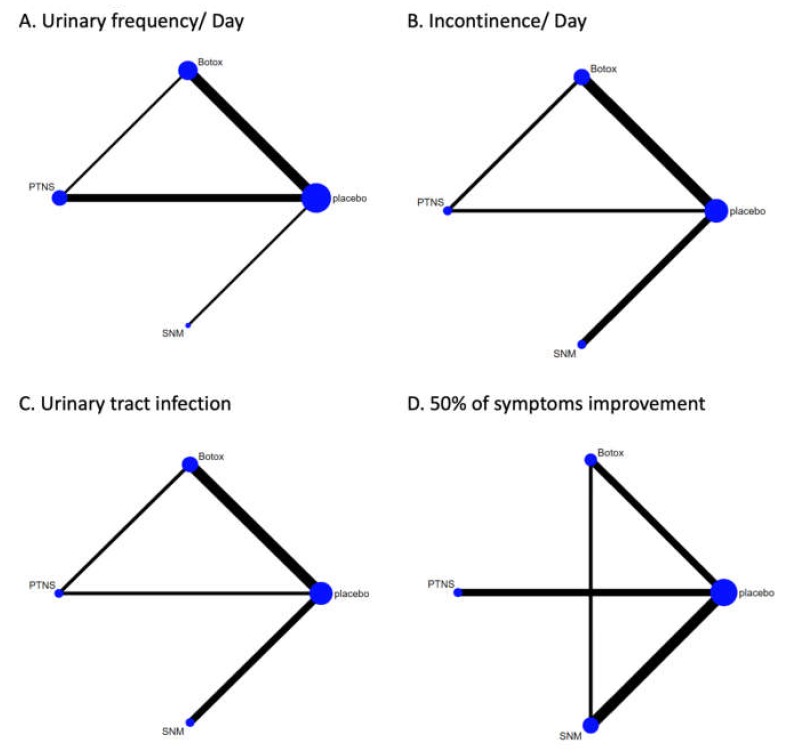
Network of treatment comparisons between OnabotulinumtoxinA (Botox), sacral neuromodulation (SNM), percutaneous tibial nerve stimulation (PTNS) in regard to (**A)** urinary frequency/day, (**B)** incontinence, (**C**) urinary tract infection, and (**D**) ≥50% of symptoms improvement. The figure was generated using R 3.3.2 software.

**Table 1 toxins-12-00128-t001:** Characteristics of the included randomized controlled trials.

Author, Year [ref.]	Trial Registration	Study Design	Participants	Exclusion Criteria	Group Sample	Follow-up (month)	Outcomes
OnabotulinumtoxinA vs. placebo							
2007 Sahai [7]	ISRCTN 16995641	Randomized, Double blinded	OAB symptoms > 6 months, refractory or intolerant to medication	Neurological disease, BOO, anticoagulant therapy, pregnancy, IC, indwelling catheter, PVR >200 mL, previous bladder surgery, UC, UTI, neuromuscular transmission disease	Cystoscopy injection OnabotulinumtoxinA 200U (*n* = 16) vs. Cystoscopy Injection with Placebo (*n* = 18)	6	Change in MMC, Urgency, UUI, urinary frequency/day, IIQ-7, UDI-6, MBC, PVR, UTI, CIC
2009 Flynn [8]	N/A	Randomized, Double blinded	OAB symptoms with UUI, refractory to medication, multiple daily incontinence and pad weight/day > 100 gm	Neurological condition, fecal incontinence or absent detrusor contraction	Cystoscopy injection OnabotulinumtoxinA 200 U/ 300 U *n* = 15) vs. Cystoscopy injection with Placebo (*n* = 7)	1.5	Incontinence, urinary frequency/ day, nocturia/ IIQ7, UDI6, pads/day, pads weight/ day, MBC, PVR, UTI, CIC
2010 Dmochowski [9]	N/A	Randomized, Double blinded	OAB symptoms with UUI > 6 months, refractory or intolerant to medication	CIC, pelvic/urological abnormalities, disease related bladder dysfunction	OnabotulinumtoxinA 50 U/100 U/150 U/200 U/300 U (*n* = 268) vs. Cystoscopy injection with Placebo (*n* = 43)	9	UUI, KHQ, UTI, CIC, PVR >200 mL, urine retention
2011 Rovner [10]	N/A	Randomized, Double blinded	OAB symptoms with UUI, refractory or intolerant to medication	Predominant SUI, pelvic or urologic abnormality or disease affect bladder function, frequent UTI, PVR >200, or VV >3000	OnabotulinumtoxinA 50 U/100 U/150 U/200 U/300 U (*n* = 268) vs. Placebo (*n* = 43)	9	UUI, urinary frequency/day, Voided volume, MBC, CIC, PVR >200 mL
2012 Denys [11]	NCT 00231491	Randomized, Double blinded	OAB syndrome and Detrusor overactivity (≥3 urgency/ 3 days, frequency), refractory or intolerant to medication	UTI, predominant SUI, PVR >150, Qmax <15, anticoagulation/ antineoplastic or exposed to OnabotulinumtoxinA	OnabotulinumtoxinA 50 U/100 U/150 U (*n* = 70) vs. placebo (*n* = 29)	6	Urgency, UUI, urinary frequency, pads/day, MBC, PVR > 50% reduction, > 75% reduction UIE, EQ-5D, IQoL, UTI, CIC
2012 Tincello [12]	ISRCTN 26091555	Randomized, Double blinded	OAB symptoms and Detrusor overactivity (frequency, ≥2 urgency /day), refractory or intolerant to medication	SUI, neurologic disease, voiding dysfunction or contraindicated to OnabotulinumtoxinA	OnabotulinumtoxinA 200 U (*n* = 122) vs. placebo (*n* = 118)	3	Incontinence, urgency, urinary frequency/day, IQoL, UTI, CIC
2013 Chapple [13]	NCT 00910520	Randomized, Double blinded	OAB syndrome with UUI, refractory or intolerant to medication in the past 12 months	Previous OnabotulinumtoxinA treatment, with neurologic reason, predominance of SUI and pelvic/ urologic abnormalities, bladder surgery or disease affect bladder function	OnabotulinumtoxinA 100 U (*n* = 277) vs. placebo (*n* = 271)	6	Incontinence, urgency, UUI, urinary frequency/day, nocturia, continent, PVR, > 50% reduction UIE, ICIQ-SF, IUSS, IQoL, UTI, CIC,
2011 Dowson [14]	ISRCTN 57577615	Randomized, Double blinded	Bladder oversensitivity, refractory or intolerant to medication	Pregnancy, breast feeding, IC, neurological condition, BOO, indwelling catheter, previous bladder surgery, previous OnabotulinumtoxinA treatment, anticoagulation agent use	OnabotulinumtoxinA 100 U (*n* = 10) vs. placebo (*n* = 11)	3	Urinary frequency/day, Urgency, UUI, IIQ-7, UDI-6, PPBC, MBC, UTI, CIC
2013 Nitti [15]	NCT 00910845	Randomized, Double blinded	OAB syndrome, refractory or intolerant to medication	Predominance of SUI	OnabotulinumtoxinA 100 U (*n* = 278) vs. placebo (*n* = 272)	3	Incontinence, urgency, urinary frequency/day, nocturia, PVR, UUI, I-QoL, KHQ, UTI, CIC
OnabotulinumtoxinA vs. PTNS							
2017 Sherif [16]	N/A	Randomized	OAB symptoms, refractory or intolerant to medication	Nerve damage, pregnant, pacemaker, defibrillator, UTI, coagulopathy, BOO, neurogenic bladder, previous RT or bladder cancer, s/p incontinence surgery	OnabotulinumtoxinA 100 U (*n* = 30) vs. PTNS (*n* = 30)	9	Incontinence, urgency, urinary frequency/day, nocturia, OABSS, QoL, frequency, nocturia, PVR, Urgency scale, MBC, UTI, CIC
PTNS vs. placebo							
2010 Finazzi-Agro [17]	N/A	Randomized, Double blinded	female, UI with detrusor overactivity incontinence, refractory or intolerant to medication	Pregnancy or plan / UTI, fistula, stone, Interstitial cystitis, DM, pacemaker/ defibrillator	PTNS (*n* = 17) vs. placebo (*n* = 15)	3	Incontinence, urinary frequency/day, nocturia, >50% reduction UIE
2010 Peters [18]	N/A	Randomized, Double blinded	OAB syndrome (OAB-q ≥4, voiding ≥10/day), refractory or intolerant to medication	Pregnant or plan/ neurogenic bladder/ previous use of OnabotulinumtoxinA / pacemaker/ defibrillator/ UTI/ use of TENS	PTNS (*n* = 103) vs. placebo (*n* = 105)	3	urinary frequency/day, nocturia, OAB-qSF, SF-36, GRA, voiding volume, UUI
2016 Scaldazza [19]	N/A	Randomized	Female with OAB syndrome	SUI, UTI, neurological disease, bladder stone, POP, pregnancy, DM, anti-incontinence surgery, pelvic tumor, radiation	PTNS (*n* = 30) vs. placebo (*n* = 30)	3	urinary frequency/day, voiding volume, nocturia, OAB-qSF, PPIUS, PGI-I >50% reduction UIE
SNM vs. Placebo							
1999 Schmidt [20]	N/A	Randomized	UUI, poor response to anti-cholinergic agents	Neurological condition, SUI, pelvic pain symptoms	SNM (*n* = 34) vs. delay SNM (*n* = 42)	6	Incontinence, pads/day, >50% reduction UIE, SF-36, implant revision
2000 Hassouna [21]	N/A	Randomized	Urgency/ frequency symptoms, refractory to medication	Neurological condition, SUI, pelvic pain symptoms	SNM (*n* = 25) vs. no SNM (*n* = 26)	6	Urinary frequency/day, MBC, >50% reduction UIE, implant revision, SF-36
2000 Weil [22]	N/A	Randomized	Refractory urinary urge incontinence	SUI, SCI, CVA within 6 months, DD, bleeding complication, VUR or hydronephrosis, UTI, pelvic pain	SNM (*n* = 22) vs. conservative treatment (*n* = 20)	6	Incontinence, pad use, implant revision rate, >50% reduction UIE
SNM vs. OnabotulinumtoxinA							
2016 Amundsen [23,24]	NCT 01502956	Randomized	UUI, refractory or intolerant to 1st and 2nd line therapy	Neurological disease, PVR >150	SNM (*n* = 174) vs. OnabotulinumtoxinA 200U (*n* = 190)	24	UUI, urinary incontinence, pads, nocturia urinary frequency/day, CIC, UTI, > 50% reduction UIE, Questionnaire SF, Satisfaction Questionnaire, PGI-I, Sandvik

CIC: clean intermittent catheterization; OABSS: overactive bladder symptom score; UTI: urinary tract infection; I-QoL: Incontinence Quality of Life Questionnaire; IIQ-7: Incontinence Impact Questionnaire, short form; MBC: maximal bladder capacity; SUI: Stress urinary incontinence; UDI-6: Urogenital Distress Inventory, Short Form; UUI: urge urinary incontinence; >50% reduction UIE: >50% reduction in urinary incontinence episodes; KHQ: King’s Health Questionnaire score; Questionnaire SF: Questionnaire short form; PGI-I: Patient Global Impression of Improvement; SF-36: Short Form 36 Health survey; PPIUS: Patient Perception of Intensity of Urgency Scale; OAB-qSF: Overactive bladder questionnaire short form; GRA: Global response assessment.

**Table 2 toxins-12-00128-t002:** Pairwise meta-analyses result for different endpoints.

Endpoint	Comparison	*N*	*I*^2^ (%)	*p* Value	Standard Mean Difference (95% CI)
Urinary frequency/ day	OnabotulinumtoxinA vs. Placebo	4	92	< 0.001	−0.65 (-0.24–−1.06)
	PTNS vs. OnabotulinumtoxinA	1			−1.02 (−1.55–−0.48)
	PTNS vs. Placebo	3	37.1	0.204	−0.37 (-0.03–−0.70)
	SNM vs. Placebo	1			−1.12 (-0.53–−1.71)
Urge urine incontinence	OnabotulinumtoxinA vs. Placebo	2	70.7	0.065	−0.37 (−0.05–-0.79)
Urgency Episode	OnabotulinumtoxinA vs. Placebo	4	97.6	<0.001	−0.84 (-0.08–-1.60)
Maximal	PTNS vs. Placebo	1			1.35 (0.79–1.92)
	SNM vs. Placebo	1			0.91 (0.33–1.48)
I-QoL	OnabotulinumtoxinA vs. Placebo	2	99.1	<0.001	0.98 (−0.89–2.86)
	PTNS vs. Placebo	1			0.86 (0.13–1.59)
Incontinence	OnabotulinumtoxinA vs. Placebo	3	97.8	<0.001	-0.84 (-1.62–-0.06)
	PTNS vs. OnabotulinumtoxinA	1			0.54 (0.02–1.06)
	PTNS vs. Placebo	1			-1.49 (-2.28–-0.70)
	SNM vs. Placebo	2	74.6	0.047	-2.10 (-3.07–-1.12)
≥50% Improvement	Placebo vs. OnabotulinumtoxinA	2	0.0	0.410	0.53 (0.40–0.70)
	PTNS vs. OnabotulinumtoxinA	2	0.0	0.371	0.50 (0.32–0.76)
	Placebo vs. PTNS	3	52.5	0.122	0.21 (0.07–0.61)
	SNM vs. Placebo	1			1.27 (0.87–1.87)
Urinary tract infection	OnabotulinumtoxinA vs. Placebo	8	0	0.486	2.55 (1.89–3.43)
	PTNS vs. OnabotulinumtoxinA	1			0.20 (0.01–4.34)
	SNM vs. OnabotulinumtoxinA	1			0.33 (0.19–0.56)
Clean intermittent catherization	OnabotulinumtoxinA vs. Placebo	9	0	0.786	5.95 (3.08–11.46)
	PTNS vs. OnabotulinumtoxinA	1			0.20 (0.01–4.34)
	SNM vs. OnabotulinumtoxinA	1			0.01 (0.00–0.23)

**Table 3 toxins-12-00128-t003:** Summary of results from NMA (on the lower triangle) and traditional pairwise meta-analysis (on the upper triangle).

	Placebo	OnabotulinumtoxinA	PTNS	SNM
Urinary frequency/ Day (SMD, 95% CI)				
Placebo	0	−0.65 (-0.24–-1.06)	−0.37 (-0.03–-0.70)	−1.12 (-0.53–-1.71)
OnabotulinumtoxinA	−1.72 (-1.23–-2.21)	0	−1.02 (−1.55–−0.48)	
PTNS	−0.80 (-0.15–-1.14)	−0.92 (−1.59–−0.26)	0	
SNM	−8.10 (-4.04–-12.16)	−6.38 (-2.29–-10.47)	−7.30 (−3.19–-11.41)	0
Incontinence/ Day (SMD, 95% CI)				
Placebo	0	−0.84 (−0.06–1.62)	−1.49 (−0.70–2.28)	−2.10 (-1.12–3.07)
OnabotulinumtoxinA	−1.96 (-0.92–-3.00)	0	−0.54 (-0.03–-1.06)	
PTNS	−2.05 (-0.56–-3.53)	-0.08 (−1.37–1.53)	0	
SNM	-10.96 (-8.60–-13.31)	−8.99 (−6.42–-11.57)	−8.91 (-6.12–-11.70)	0
Urinary tract infection (OR, 95% CI)				
Placebo	1	2.54 (1.89–3.44)		
OnabotulinumtoxinA	3.06 (2.26–4.15)	1	0.20 (0.01–4.35)	0.33 (0.19–0.56)
PTNS	0.57 (0.03–12.62)	0.19 (0.01–4.06)	1	
SNM	10.73 (0.39–1.38)	0.24 (0.14–0.42)	1.28 (0.06–29.29)	1
Clean intermittent catheterization (OR, 95% CI)				
Placebo	1	5.95 (3.08–11.46)		
OnabotulinumtoxinA	6.92 (3.18–15.06)	1	0.20 (0.01–4.34)	0.01 (0.00–0.23)
PTNS	1.29 (0.05–31.93)	0.19 (0.01–4.19)	1	
SNM	0.08 (0.00–1.46)	0.01 (0.00–0.19)	0.06 (0.00–4.01)	1

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
