# Peer review of "Comparing the Efficacy of OnabotulinumtoxinA, Sacral Neuromodulation, and Peripheral Tibial Nerve Stimulation as Third Line Treatment for the Management of Overactive Bladder Symptoms in Adults: Systematic Review and Network Meta-Analysis"

_toxins, 2020, doi:10.3390/toxins12020128_

Round 1
Reviewer 1 Report
The conclusion of the manuscript ‘Comparing the efficacy of onabotulinumtoxinA, sacral neuromodulation, and peripheral tibial nerve stimulation as third line treatment in the management of overactive bladder in adults: A Systematic Review and Network Meta-analysis’ is not surprising; Good quality and especially long term evidence is lacking, and one of the three treatments discussed is slightly better on selected outcomes. Not on ‘overactive bladder’ (as is announced in the title) but on ‘urinary incontinence episodes’ and ‘urinary frequency per day’. ‘Overactive bladder’ should be ‘overactive bladder syndrome’ and should be replaced everywhere in the text. The overactive bladder syndrome contains a proportion of patients with detrusor overactivity and for this only there is some evidence of a mechanism of action of the 3 studied treatments. The authors are not to blame but (outcome) research with patients in a syndrome is blurred by definition. All cohorts are mixed regarding patho-physiology; only subjective symptoms at entry are collected and subjective symptoms at outcome. If the authors state that better research -with better follow up- is needed. I would say, do not waste your time with a summary of mediocre research and start or continue the research that you consider necessary. This review is one of the many concerning this topic and including (this type of or) these studies, and I expect another systematic review and meta-analysis with these studies adding two new studies as soon as they are published…. The review is adequately done and the rules are followed. However when registries as clinicaltrials.gov are included the intended power (numbers to recruit) and the intended inclusion, exclusion as well as outcome can be scrutinized and compared with the publication of the study. This helps to judge potential bias (at entry, outcome and potential post-hoc exclusion or lost to follow up, better than done now and can be included in the manuscript. A meticulous and correct review of fuzzy research can not end in clear results and this manuscript concludes similar as the fuzzy literature review of the AUA guideline. Nothing is changed. There is no need to publish this. Would it be nice for clinicians to know? I think that the data, presented with only effect sizes for each symptom without 'harm' including e.g. costs and time & material needed, without infections other than UTI, without e.g. QuaLY's (Indeed: The included (available) manuscripts are to blame!) is too abstract for clinicians to be applicable/ practice changing. I would have appreciated a much more critical review of the published research and a genuine initiative to guide, develop and collect real necessary objective data about our (diagnosis and) management of patients with objectively demonstrated dysfunction.
Reviewer 2 Report
I think that this study is interesting and timely. It is also challenging to conduct a Network Meta-analysis, especially when studies are heterogeneous and some are of low quality.
1. Your text contains spelling mistakes (line 248 compied), grammatical errors (Table 1 "refractory to anticholinergic") and statements which are very difficult to undertand. I advise a re-write with the help of an expert in writing Scientific English. Please be consistent and observe journal house style. You have used the terms anticholinergic and antimuscarinic interchangeably. Likewise there is inconsistency in the text, figures and tables in the use of Botox (please avoid), Botulinum toxin A and Onabotulinum toxin.
2. In your methods you seem to confuse study quality, ROB and GRADE. Your results section contains information on ROB but little on other aspects of study quality. In addition your ROB results (Figure 2) paints a very positive view of the included studies, most were graded as "low ROB". This is very far removed from the ROB reported in the majoirty of Cochrane reports. Have you applied the ROB 2.0 tool correctly?
3. In Figure 1 the numer of studies given after the removal of duplicates is the sum of the number of studies identified in Pubmed (1940) and Embase (5722) 1940+5722= 7662. Either this means there were no duplicates (impossible) or you have a basic error in your search strategy.
Round 2
Reviewer 1 Report
The editor has allowed to revise the manuscript ‘Comparing ….onabotulinumtoxinA, sacral neuromodulation, and peripheral tibial nerve stimulation …A Systematic Review and Network Meta-analysis’. The response to the reviewer letter does not at all address the reviewers questions and suggestions, however many words are changed in the text, the predominance of the earlier review is still valid. Also I have asked to replace ‘Overactive bladder (or OAB)’ with ‘overactive bladder syndrome (e.g. OAB-s)’ and although the title is changed the text is not. I do not repeat my earlier review but hope that my concerns are better addressed at second look. I am very concerned that most of the studies carry a high or unknown risk of bias (esp. a high chance of post-hoc exclusion, and or adaptation of ‘primary’ outcome and or p-hunting); a better reporting of the available pre-trial evidence /documents (protocol with primary endopints) e.g. ClinicalTrialsGov. is needed to convince me of the contrary. Replying to the covering letter: a (n overstated) conclusion based on a nice collage of biased research is never helpful for clinicians! A conclusion is impossible without the ‘further research’ that is ‘automatically’ written in the conclusion. To clarify my concerns again, one example: the Rosetta (not ‘rosseta’) trial has included >30% of patients in both arms without overactive detrusor (see the table in the article). There is no known mechanism of action of both therapies for frequent voiding without overactive detrusor…. ..This violates not only clinical principles but also ethical principles and makes this (but also other) research already biased by inclusion.
My recommendation is to publish the review, but with a very critical conclusion about the quality of the research and the quality of the evidence. Not ‘further research must be done’ but ‘much better research must be done’ (with better objective and pre-specified inclusion and outcomes).
Reviewer 2 Report
Thank you for clarification of the search strategy and results and for other changes which have improved your manuscript.
Round 3
Reviewer 1 Report
The manuscript is somewhat adapted; after my two reviews. The word ‘symptoms’ is now added with to about 50% of the abbreviations ‘OAB’. I have asked to use the correct paraphrase from the standard that the authors refer to themselves: I copy from that document: ……[Abrams et al:] ‘[par] 1.7.2. Symptom syndromes suggestive of lower urinary tract dysfunction. In clinical practice, empirical diagnoses are often used as the basis for initial management after assessing the individual’s lower urinary tract symptoms, physical findings and the results of urinalysis and other indicated investigations. Urgency, with or without urge incontinence, usually with frequency and nocturia, can be described as the overactive bladder syndrome, urge syndrome or urgency-frequency syndrome. (NEW)’….. [end of citation] I assume the authors see the reason to, now meticulously, adapt the manuscript.
I copy also from the submitted manuscript: startling from line 217, but in a slightly adapted version, as a suggestion:
[adapted text] ……. Different protocols were also used for PTNS. Second, the variable or ambiguous and potentially post-hoc definitions for the outcome parameters used in each study made it difficult to unify and compare the treatment results between the studies. However, because of the heterogeneity of enrolled patients, existing of high‐risk bias and lacking long term outcomes comparison, more high‐quality studies are necessary to clarify this benefit and risk of the current available 3rd line therapy for overactive bladder symptoms. A strength of the current study was that it collected 17 randomized controlled trials published in English over three decades and used network meta‐analysis to compare the three existing therapeutic options.
4 Conclusions
The results revealed that all three modalities were efficacious in managing adult patients with the OAB syndrome, and all were better than a placebo on the specific symptom reported to be the outcome of the study. This review shows that at 12 weeks follow‐up, SNM yielded the greatest reduction in urinary incontinence episodes and urinary frequency/day. As there is a lack of head to head comparison studies among SNM, PTNS and OnabotulinumtoxinA for the treatment of adult OAB symptoms, the current network meta-analysis represents the best available evidence to estimate the relative potential of these three treatment modalities. [end of adapted text]. I ask the authors again to consider a conclusion, based on the results of their analysis.
Furthermore I copy the abstract; also slightly adapted….
[adapted text:] The American Urological Association guidelines for the management of non neurogenic overactive bladder (OAB) recommend the use of OnabotulinumtoxinA, sacral neuromodulation (SNM) and peripheral tibial nerve stimulation (PTNS) as third line treatment options with no treatment hierarchy. The current study used network meta‐analysis to compare the efficacy of these three modalities for managing adult patients with the OAB syndrome. We performed systematic literature searches of several databases from Jan 1995 to Sep 2019 with language restricted to English. All randomized control trials that compared any dose of OnabotulinumtoxinA, SNM and PTNS with each other or a placebo for the management of adult OAB syndrome were included in the study. Overall, 17 randomized control trials, with a follow up of 3-6 months in the predominance of trials (range 1,5-24 months), were included for analysis. For each trial-outcome, the results were reported as an average number of episodes of the reported outcome at baseline. Compared with the placebo, all three treatments were more efficacious for the selected outcome parameter. OnabotulinumtoxinA resulted in a higher number of complications, including urinary tract infection and urine retention. Compared with OnabotulinumtoxinA and PTNS, SNM resulted in the greatest reduction in urinary incontinence episodes and voiding frequency. [end of adapted text]
I must react on this sentence in the letter: [citation] ‘Detrusor overactivity on UDS is not a requisite component for diagnosis of overactive bladder.’ Even my youngest junior medical student understands the difference between ‘symptoms’ and ‘diseases’ (dysfunctions) and or ‘diagnosis’. None of these students would tell me that ‘…..it is not a requisite to make an x-ray photograph of the thorax to know that the patient is coughing and sneezing.’
Round 4
Reviewer 1 Report
none